# Estimating the effect of moving meat-free products to the meat aisle on sales of meat and meat-free products: A non-randomised controlled intervention study in a large UK supermarket chain

**Carmen Piernas**[1]*, **Brian Cook**[1], **Richard Stevens**[1], **Cristina Stewart**[1], **Jennifer Hollowell**[1], **Peter Scarborough**[2], **Susan A. Jebb**[1]

1 Nuffield Department of Primary Care Health Sciences, University of Oxford, Oxford, United Kingdom,
2 Nuffield Department of Population Health, University of Oxford, Oxford, United Kingdom

* carmen.piernas-sanchez@phc.ox.ac.uk

## Abstract

### Background

Reducing meat consumption could bring health and environmental benefits, but there is little research to date on effective interventions to achieve this. A non-randomised controlled intervention study was used to evaluate whether prominent positioning of meat-free products in the meat aisle was associated with a change in weekly mean sales of meat and meat-free products.

### Methods and findings

Weekly sales data were obtained from 108 stores: 20 intervention stores that moved a selection of 26 meat-free products into a newly created meat-free bay within the meat aisle and 88 matched control stores. The primary outcome analysis used a hierarchical negative binomial model to compare changes in weekly sales (units) of meat products sold in intervention versus control stores during the main intervention period (Phase I: February 2019 to April 2019). Interrupted time series analysis was also used to evaluate the effects of the Phase I intervention. Moreover, 8 of the 20 stores enhanced the intervention from August 2019 onwards (Phase II intervention) by adding a second bay of meat-free products into the meat aisle, which was evaluated following the same analytical methods.

During the Phase I intervention, sales of meat products (units/store/week) decreased in intervention (approximately −6%) and control stores (−5%) without significant differences (incidence rate ratio [IRR] 1.01 [95% CI 0.95–1.07]. Sales of meat-free products increased significantly more in the intervention (+31%) compared to the control stores (+6%; IRR 1.43 [95% CI 1.30–1.57]), mostly due to increased sales of meat-free burgers, mince, and sausages. Consistent results were observed in interrupted time series analyses where the effect of the Phase II intervention was significant in intervention versus control stores.

**Data Availability Statement:** This research was conducted according to a framework collaboration agreement between the University of Oxford and the food retailer. Access to the study dataset by external researchers is not permitted without the

expressed written consent of the retailer as this is defined as confidential information in the agreement.

**Funding:** This analysis was funded by the Wellcome Trust, Our Planet Our Health (Livestock, Environment and People – LEAP), award number 205212/Z/16/Z. SAJ is a National Institute of Health Research (NIHR) senior investigator and is funded by NIHR Oxford Biomedical Research Centre. SAJ and CP are funded by the Oxford and Thames Valley NIHR Applied Research Centre. The retail partner funded the cost of the intervention itself and extracted the data for analysis. They selected the intervention and control stores but had no further part in developing the protocol, conducting the analysis, or drafting the paper. The principal funder of the study and the participating retailer had no role in study design, data collection, data analysis, data interpretation, preparation of the manuscript or decision to publish.

**Competing interests:** The authors have declared no competing interests exist.

## Conclusions

Prominent positioning of meat-free products into the meat aisle in a supermarket was not effective in reducing sales of meat products, but successfully increased sales of meat-free alternatives in the longer term.

A preregistered protocol (https://osf.io/qmz3a/) was completed and fully available before data analysis.

---

## Author summary

### Why was this study done?

- Reducing meat consumption could bring health and environmental benefits, but there is limited evidence of interventions that may be effective.

- Physical environments within which food choices are made can exert a significant influence on food selection. Supermarkets account for a large proportion of all United Kingdom retail grocery sales, and interventions to reduce meat selection store have the potential to reach a large number of consumers. But evidence on the effectiveness of in-store "nudges" to shift meat purchasing behaviours is lacking.

- We assessed the immediate and longer-term effects of repositioning meat-free products (plant-based mince, burgers, meatballs, and sausages) into the meat aisle in a major UK supermarket on the purchases of meat and meat-free products.

### What did the researchers do and find?

- Prominent positioning of meat-free products in the meat aisle did not result in decreased sales of equivalent meat products (the primary outcome measure).

- However, the intervention led to a significant increase in sales of meat-free products, which was sustained over time. Intensification of the intervention in the Phase II period further increased sales of meat-free products, with no impact on sales of meat products.

- A post hoc analysis by store affluence suggested that the effects on sales of meat-free products were of greater magnitude in stores located in areas of average or below average affluence.

### What do these findings mean?

- Redesigning microenvironments in grocery stores could help shift habitual purchasing behaviours to reduce the demand for meat.

- The limited effectiveness of this positioning intervention in meeting the primary goal, reduced meat consumption, suggests that it may be necessary to make structural changes that explicitly target a reduction in meat purchases to achieve global targets that are compatible with human health and environmental sustainability.

## Introduction

Globally, the consumption of meat is rising, driven by population growth and increasing economic development. [1] There is evidence that consuming red and processed meat is associated with adverse health effects, including heart disease and cancer [2,3]. Furthermore, livestock production negatively affects the natural environment through greenhouse gas emissions and other outcomes [4]. Reducing meat consumption could bring health and environmental benefits, but there is little research to date on effective interventions to achieve this [5].

It is now widely recognised that the physical environments within which food choices are made can exert significant influence on food selection and are major determinants of dietary behaviour and obesity [6–8]. With regard to supermarkets, evidence suggests that current practices tend to encourage purchasing of energy-dense processed products [9,10]. Still, supermarkets may be a particularly promising setting for interventions aiming to influence consumer demand for meat because they account for approximately 87% of all UK retail grocery sales and their potential to reach a large number of consumers [11]. However, the evidence on the effectiveness of in-store "nudges" to shift meat purchasing behaviours is lacking. A systematic review of choice architecture interventions to reduce meat consumption found that 4 interventions that repositioned meat products to be less prominent at point of purchase showed promising reductions in the demand for meat. However, these studies were all conducted in buffet/canteen settings with restricted choices and food purchased for immediate consumption, with none in supermarket settings [12].

Systematic reviews that have drawn on broader evidence relating to interventions to shape food purchasing have found moderate- to low-quality evidence that product placement strategies (e.g., availability or prominent positioning (end of aisles, checkouts, or at eye level)) implemented in food retail environments can influence dietary behaviours [13,14]. Also, "cross-category merchandising" (where complementary products are colocated in a store) is an established sales mechanism to increase sales [15,16]. Research has shown that the provision of meat-free alternatives can help people to reduce their consumption of meat [17]; hence, positioning meat-free alternative products in store alongside meat products presents a potential opportunity to encourage switching from meat to meat-free alternatives.

This study aimed to evaluate an in-store positioning intervention in which a major UK supermarket moved 26 meat-free products (plant-based mince, burgers, meatballs, and sausages) into the meat aisle. Through a collaboration agreement with the supermarket, we obtained aggregated weekly store-level sales data, comprising sales of meat and meat-free products for 20 intervention stores and 88 matched control stores. Using this large dataset of supermarket sales, we aimed to assess the immediate and longer-term effects of repositioning meat-free products in the meat aisle on the purchases of meat and meat-free products.

## Methods

### Data source

Data on store-level weekly sales (units purchased and £) of meat and meat-free products were obtained from August 2017 to December 2019 (123 weeks) for 108 stores located across the UK (20 intervention and 88 matched control stores), resulting in 13,284 total store weeks of data. No stores were excluded from primary or secondary analyses due to the overall completeness of the data. For one store which had missing data for 2 non-consecutive weeks in August and September 2018 (0·01% store weeks), values were imputed using the average of each month. The study used only aggregate sales data to evaluate actions implemented by a supermarket and was exempt from ethical review and approval.

## Store selection and matching

The retail partner's finance and data teams determined the criteria for selecting the intervention and control stores. Intervention sites were all stores where there was unproductive space in the meat aisle. In other words, the meat sales did not justify the volume of inventory or number of product bays. For this study, 20 stores were selected based on the retailer's estimates of the opportunity cost of removing meat products, as well as operational considerations. The stores were selected to have a broad spread with regard to the following characteristics: affluence (based on top 2 Acorn consumer classification categories within a 10-minute drive) [18]; sales of meat-free products over a 12-week period ending September 25, 2018; and store size (large (>50,000 sq ft), medium (35 to 50,000 sq ft), and small (<35,000 sq ft).

The retail partner used proprietary analytics software to select 88 unique control stores with similar purchasing patterns to the intervention stores based on individual-level purchasing data captured by their loyalty card scheme. For this study, the retailer identified up to 5 matched control stores for each intervention store. This was based on patterns of customer purchasing of meat-free alternatives over the 52 weeks preceding the intervention period as well as data on the store format, visits per customer, spend per visit, and customer affluence. A small number of control stores was initially matched to more than 1 intervention store. In these instances, we manually selected only 1 control per intervention store. Therefore, of the 20 intervention stores, 9 had 5 uniquely matched controls, 10 had 4 uniquely matched controls, and 1 had 3 uniquely matched control stores. The final sample of 108 stores included in this analysis were representative of the retail partner store population, with stores distributed across all UK regions, including small, medium, and large supermarket sites.

## Intervention

The intervention was implemented in 20 stores on the week commencing January 27, 2019, for 12 weeks. In the control stores, meat-free products remained where originally located (usually in their own meat-free section outside the meat aisle) without changes to their position. The intervention consisted of moving 26 uniquely barcoded meat-free products within 4 meat-free categories (50% sausages, 30% burgers, 12% meatballs, and 8% mince) from their usual section into a newly created meat-free bay within the meat aisle. This new bay of meat-free products covered the whole bay, including top, middle, and bottom shelves. This meat-free section replaced a bay where either meat was previously sold (16 stores) or where the bay previously contained chilled condiments (3 stores) or was unused (1 store). The meat-free bay was the end bay in the meat aisle in 13 stores and in the middle of the meat aisle in 7 stores.

Additionally, the meat-free bay had a promotional header board above it in most stores (with the words "Meat Alternatives" and a meat-free burger image) and 2 point of sale displays (aisle fins) that included the following text, developed by the retail partner:

- Simple swaps—More delicious meat alternatives can be found across the store

- Plant power—Easy switches for an alternative source of protein.

The intervention was initially planned by the retail partner for 12 weeks until April 2019 (referred to as intervention Phase I). However, 12 of the 20 intervention stores continued the Phase I intervention until December 2019. For the remaining 8 intervention stores, the retailer intensified the intervention (referred to as intervention Phase II) in the week commencing July 28, 2019 by adding the entire range of meat-free products into a second meat-free bay within the meat aisle. This second bay of meat-free products included additional meat-free products (e.g., falafels, sausage rolls, tofu, and slices), which were stocked alongside the

previous 26 products included in the Phase I intervention. These 8 stores were selected by the retailer based on each stores having sufficient meat-free products in stock for the second bay. The Phase II intervention continued in these 8 stores until December 2019.

## Outcome measures

The primary outcome measure was average total weekly sales (units) of total meat products (aggregated measure of mince, burgers, meatballs, and sausages). Secondary outcomes included average total weekly sales (units) of total meat-free products (aggregated measure of meat-free mince, burgers, meatballs, and sausages); total weekly sales (£) of total meat products and total meat-free products; total weekly sales (units and £) of individual categories (mince, burgers, meatballs, and sausages) of meat products and meat-free products; total weekly sales (units and £) of other meat-free products not included above (e.g., tofu and falafel); and total weekly sales (units and £) of different aggregated product categories that were chosen as control products, including fish, nondairy milk, vegetables, fruit, and personal care products.

## Other measures

The retailer also provided data on demographic characteristics relating to the surrounding population. These data were originally derived using the Acorn categorisation system, which uses postcode data as well as commercial and governmental data to classify geographical areas across the UK [18]. These measures were included as covariates in the analyses: affluence coded as average, >average, <average; age group coded as older versus younger; ethnicity coded as White, Asian, or Other; and area density coded as urban, more urban, or less urban.

## Statistical analysis

A preregistered protocol (https://osf.io/qmz3a/) was completed and fully available from April 2019 before obtaining data for analyses (S1 Appendix). A more detailed statistical analysis plan was completed in January 2020 after data cleaning but before data analysis (S2 Appendix). Following the conduct and evaluation of a natural experiment, a power analysis was not conducted, and the final number of trial stores with matched control stores was chosen by the retailer.

For the primary analysis we used 2 prespecified 12-week time periods: (a) a pre-intervention baseline period (week commencing September 2, 2018 to November 18, 2018); and (b) a Phase I intervention period (week commencing February 3, 2019 to April 21, 2019), which excludes the week of January 27, 2019 to allow for the intervention to be fully implemented across all stores. The pre-intervention baseline period and the intervention period were defined a priori with the retail partner to avoid months where meat and meat-free purchases are known to be atypical (e.g., Christmas, January, and summer).

We investigated differences in store demographic characteristics between intervention versus control stores using chi-squared tests. We also investigated differences in aggregated store-level weekly sales (units and £) between intervention versus control stores over the pre-intervention baseline period (September 2, 2018 to November 18, 2018) as well as in January 2018 and January 2019 using Student t tests.

The primary outcome analysis evaluated the effect of the Phase I intervention by comparing changes in items/units sold per week of total meat products between intervention versus control stores during the Phase I intervention period (week commencing February 3, 2019 to April 21, 2019). We used a hierarchical negative binomial model (to account for the non-normal distribution of the outcome data—units sold/week) to obtain incidence rate ratios (IRRs), with fixed effects adjustment for store affluence, store age group, store ethnicity, store area,

average units sold per week over the 12-week pre-intervention baseline period (week commencing September 2, 2018 to week commencing November 18, 2018), and a random effects term for matching group. Analyses of all the other secondary outcomes followed the same approach except for using a hierarchical normal model where the outcomes were sales (£) and followed a normal distribution.

In a prespecified secondary analysis, we conducted interrupted time series analyses on all available data from August 1, 2017 to week commencing December 1, 2019 using sales (units, £) of total meat and meat-free products [19]. For evaluation of the Phase I intervention, we plotted (a) the average sales of meat and meat-free products across all 20 intervention stores; and (b) the average sales of meat and meat-free products across all 88 control stores, together with fitted linear trend lines before and after the Phase I intervention began in the week commencing February 3, 2019. To assess whether differences visible in this graph were statistically significant between intervention and control stores, we used a difference-in-difference approach, calculating the mean difference in sales (units or £) at each week between intervention and control group, and testing whether this time series of differences changed after intervention time using a linear regression model. We used a Chow-type test for any difference (in either intercept or slope, or both) after versus before and used Newey–West standard errors with lag 2 to allow for autocorrelation in the time series. In sensitivity analyses, we tested the effect of controlling for seasonality. We used the same approach to evaluate the Phase II intervention, but we only used data from the 20 intervention stores (12 Phase I and 8 Phase II stores) from the of February 3, 2019 to December 1, 2019. In addition to the hierarchical mixed models, we used interrupted time series to plot (a) the average meat and meat-free sales across the 12 Phase I intervention stores; and (b) the average meat and meat-free sales across the 8 Phase II stores, together with fitted linear trend lines before and after the Phase II intervention began in the week commencing July 28, 2019.

In a post hoc sensitivity analysis, we repeated the same hierarchical mixed models for the primary outcome (unit sales) adjusting for an alternative baseline period (week commencing February 4, 2018 to April 22, 2018), which matched the intervention period (February 3, 2019 to April 21, 2019). In another post hoc exploratory analysis, we used interaction terms (store affluence * intervention) in the hierarchical negative binomial model described above to compare the effect of the intervention across store affluence groups. We used R 3.6.0 to produce graphics and Stata version 16 for the remaining statistical tests. All statistical tests were conducted at a 5% significance level.

### Fidelity evaluations

We trained and employed community researchers to carry out in-store fidelity evaluations of all Phase I and Phase II intervention stores and a sample of control stores on 2 unannounced occasions throughout the intervention periods (1 week day and 1 weekend day). These visits aimed to assess whether the intervention was implemented as planned and collected information using a survey and photographs of the participating stores.

### Results

This analysis included data from 108 stores, the majority of which were located in areas of average (49%) or above average affluence (39%) and below average urbanisation (61%). Approximately half of the stores were located in areas where the main ethnicity was White (51%), followed by Asian (41%). There were no significant differences in any of these characteristics between all intervention stores compared to control (Table 1), neither between

**Table 1. Demographic store characteristics relating to the surrounding population.**

| | Total stores n = 108 | | Intervention stores n = 20 | | Control stores n = 88 | | χ² test |
|---|---|---|---|---|---|---|---|
| | *n* | % | *n* | % | *n* | % | *P* value |
| *Store size* | | | | | | | |
| Small | 49 | 45 | 9 | 45 | 40 | 45 | 0.994 |
| Medium | 33 | 31 | 6 | 30 | 27 | 31 | |
| Large | 26 | 24 | 5 | 25 | 21 | 24 | |
| *Age* | | | | | | | |
| Older | 52 | 48 | 11 | 55 | 41 | 47 | 0.497 |
| Younger | 56 | 52 | 9 | 45 | 47 | 53 | |
| *Affluence* | | | | | | | |
| Less than average | 13 | 12 | 2 | 10 | 11 | 13 | 0.953 |
| Average | 53 | 49 | 10 | 50 | 43 | 49 | |
| More than average | 42 | 39 | 8 | 40 | 34 | 39 | |
| *Ethnicity* | | | | | | | |
| White | 55 | 51 | 11 | 55 | 44 | 50 | 0.327 |
| Asian | 44 | 41 | 9 | 45 | 35 | 40 | |
| Other | 9 | 8 | 0 | 0 | 9 | 10 | |
| *Area density* | | | | | | | |
| Less urban | 66 | 61 | 12 | 60 | 54 | 61 | 0.510 |
| Average urban | 40 | 37 | 7 | 35 | 33 | 38 | |
| More urban | 2 | 2 | 1 | 5 | 1 | 1 | |

intervention stores which implemented the Phase I intervention (*n* = 12) compared to those that implemented the Phase II intervention (*n* = 8, Table A in S3 Appendix).

A trend analysis over the months preceding the implementation of the intervention (August 2017 to January 2019) showed that sales of meat products and equivalent meat-free products followed similar trends in both the intervention and control stores (Fig A in S3 Appendix). The selected meat products (mince, burgers, meatballs, and sausages) accounted for up to 14 times more sales than their equivalent meat-free products. There were clear seasonal changes in meat and meat-free products in the December to January period, with another peak of sales of meat products around April to May. There were no significant differences in average weekly sales of meat or equivalent meat-free products over the 12 week pre-intervention baseline period (week commencing September 2, 2018 to week commencing November 18, 2018) or the months of January 2018 and January 2019, between intervention and control stores (Table B in S3 Appendix).

After the implementation of the intervention Phase I (February to April 2019), sales of meat products (units per store per week) decreased in both intervention (approximately −6%) and control stores (−5%), but these changes were not significantly different (IRR 1.01 [95% CI 0.95 to 1.07], Fig 1, Table C in S3 Appendix). Changes in sales (units per store per week) of individual categories of mince, burgers, meatballs, or sausages were not significantly different either in the intervention stores compared to the control stores, nor were there differences when the outcomes were expressed as £ sales of meat products (Table C in S3 Appendix).

Sales of equivalent meat-free products (units per store per week) increased significantly more in the intervention stores (approximately +31%) compared to the control stores (+6%, IRR 1.43 [95% CI 1.30 to 1.57]. This change was mostly due to increased sales of meat-free burgers, mince, and sausages (Fig 1, Table C in S3 Appendix). Significant changes were also

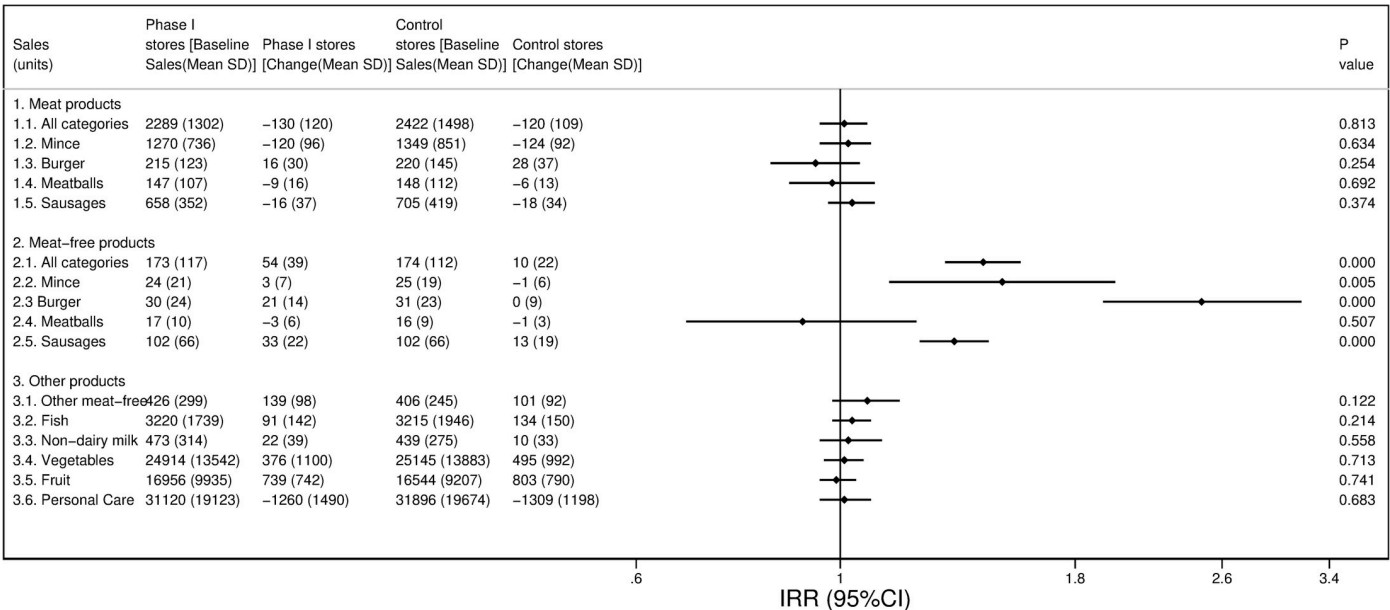

**Fig 1. Average baseline sales (units/items sold) per store per week and comparison of changes between intervention and control stores over the Phase I intervention period.** *IRRs (95% CI) comparing changes in intervention stores vs. control stores using hierarchical negative binomial models with fixed effect adjustment for store affluence, store age group, store ethnicity, store area and average units sold per week in 12-week pre-intervention period (week commencing September 2, 2018 to week commencing November 18, 2018), and a random effects term for matching group. IRR, incidence rate ratio.

observed when the outcomes were expressed as £ sales, especially for sales of meat-free burgers and sausages (Table C in S3 Appendix). Changes in sales (units per store per week) of other meat-free products or products that were chosen as control products (including fish, nondairy milk, vegetables, fruit, and personal care) did not differ significantly between the intervention and control stores (Fig 1, Table D in S3 Appendix), but there were significant increases in sales (£ per store per week) of other meat-free products (£72, 95% CI 75 to 87) and nondairy milk (£17, 95% CI 4 to 30). A post hoc sensitivity analysis was performed on sales (units per store per week) of meat and meat-free products adjusting for an alternative baseline period, which matched more closely the intervention period, with both analyses showing consistent results (Table E in S3 Appendix).

Interrupted time series analyses of the main intervention Phase I (February to April 2019) showed consistent results (Fig 2, Figs B and C in S3 Appendix), with a decreasing trend in baseline sales (units and £ per store per week) of meat products before and after the intervention and no significant difference in the trend in sales of meat products after implementation ($P = 0.7$ for differences between intervention versus control stores). For meat-free products, there was a modest declining trend in sales before intervention implementation, but a significant increase in sales of meat-free products after the intervention ($P < 0.001$ for differences between intervention versus control stores). This was followed by an increasing secular trend in meat-free sales in both intervention and control stores. These results were robust in sensitivity analyses that adjusted for seasonal changes, e.g., those observed over the months of December and January (Fig D in S3 Appendix).

Hierarchical mixed methods (Tables F and G in S3 Appendix) as well as interrupted time series analyses (Fig 3) were also used to evaluate the Phase II intervention, where 8 of the 20 original intervention stores moved the entire meat-free bay into the meat aisle, while the remaining 12 stores continued as in the Phase I intervention. In interrupted time series

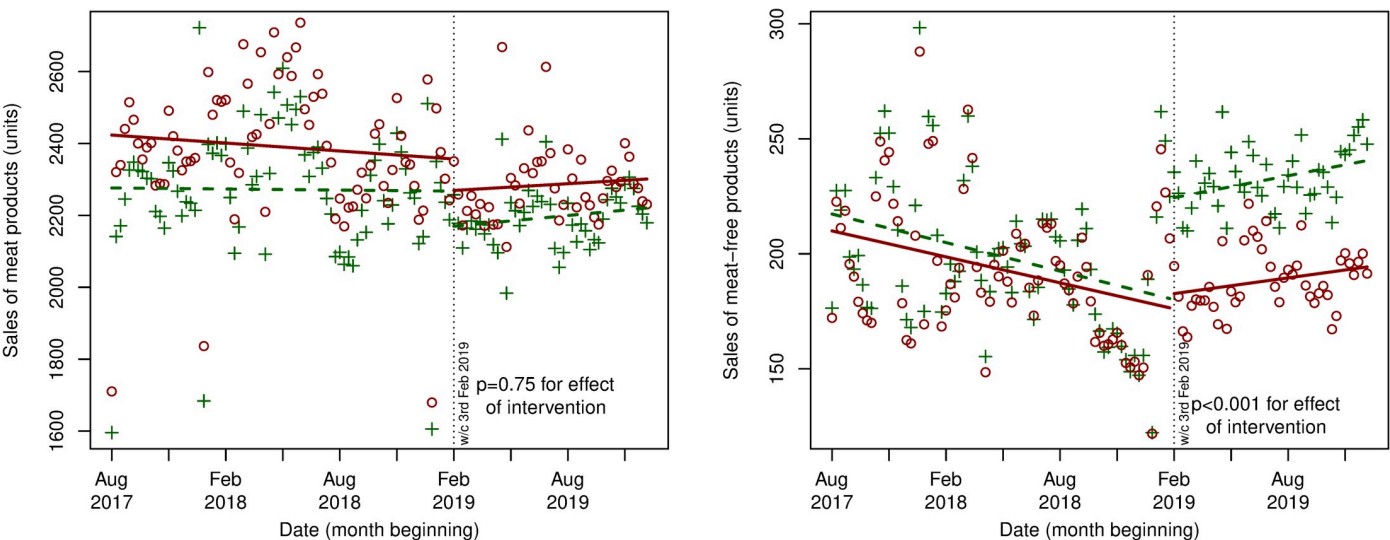

**Fig 2. Interrupted time series analysis showing level and trend changes in sales (units) of meat and meat-free products before and after Phase I intervention (week commencing February 3, 2019) in intervention stores (*n* = 20) and control stores (*n* = 88).** \*Red dots denote control stores, and green crosses denote intervention stores.

analyses, there was a declining trend in sales (units per store per week) of meat products before the intervention in both Phase I and Phase II stores and a small significant level change in meat products after implementation ($P < 0.001$ for differences between Phase I versus Phase II stores), followed by a slightly increasing trend in meat products after the intervention in both Phase I and Phase II stores. For meat-free products, there was an increasing trend in sales (units per store per week) in the Phase II stores and a significant change in meat-free products after implementation of Phase II ($P < 0.001$ for differences between Phase I versus Phase II stores) followed by an increasing trend in both Phase I and Phase II stores.

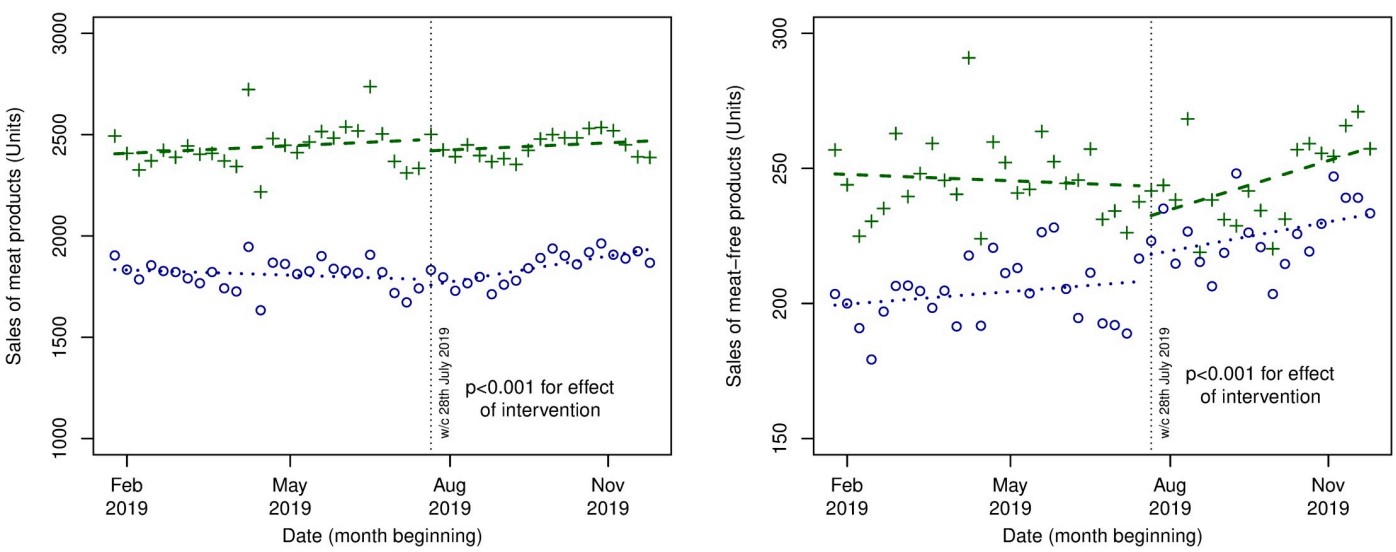

**Fig 3. Interrupted time series analysis showing level and trend changes in sales (units) of meat and meat-free products before and after Phase II intervention period (week commencing July 28, 2019) in Phase I stores (*n* = 12) and Phase II stores (*n* = 8).** \*Green crosses denote intervention Phase I stores, and blue dots denote intervention Phase II stores.

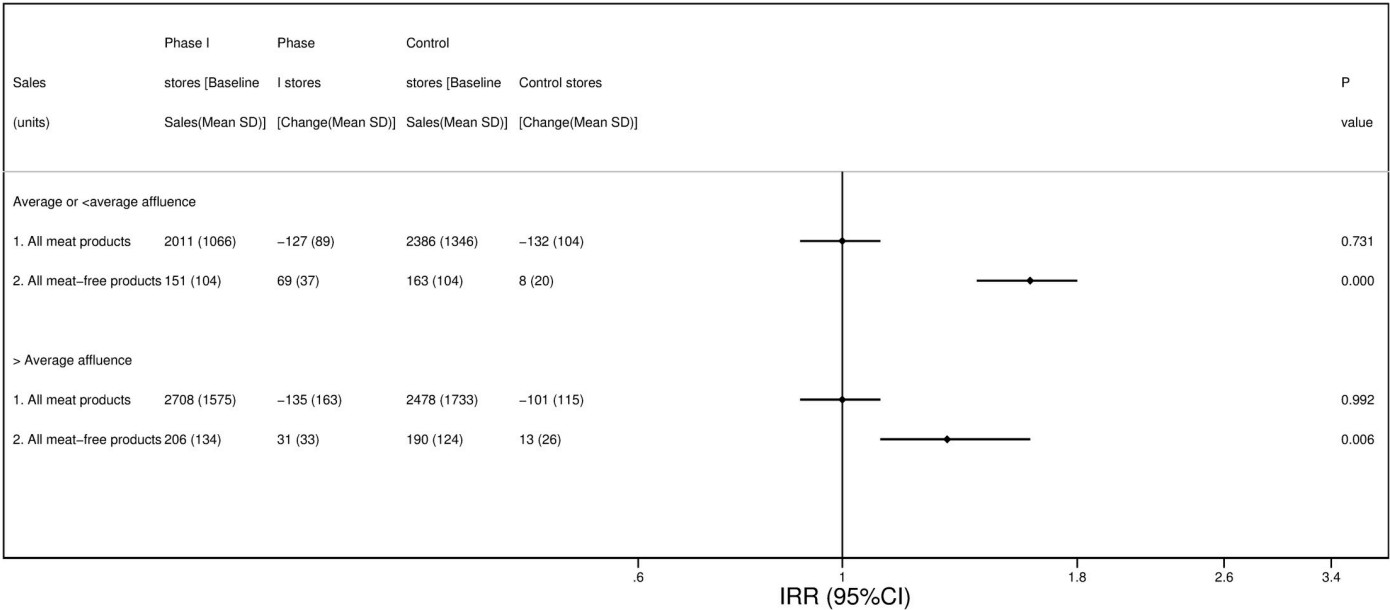

| Sales (units) | Phase I stores [Baseline Sales(Mean SD)] | Phase I stores [Change(Mean SD)] | Control stores [Baseline Sales(Mean SD)] | Control stores [Change(Mean SD)] | | P value |
|---|---|---|---|---|---|---|
| **Average or <average affluence** | | | | | | |
| 1. All meat products | 2011 (1066) | −127 (89) | 2386 (1346) | −132 (104) | | 0.731 |
| 2. All meat–free products | 151 (104) | 69 (37) | 163 (104) | 8 (20) | | 0.000 |
| **> Average affluence** | | | | | | |
| 1. All meat products | 2708 (1575) | −135 (163) | 2478 (1733) | −101 (115) | | 0.992 |
| 2. All meat–free products | 206 (134) | 31 (33) | 190 (124) | 13 (26) | | 0.006 |

**Fig 4. Average baseline sales (units/ items sold) per store per week and comparison of changes between intervention and control stores over the Phase I intervention period by store affluence.** *Changes in intervention vs. control stores were compared using hierarchical negative binomial models with fixed effect adjustment for store affluence, store age group, store ethnicity, store area and average units sold per week in 12-week pre-intervention period (week commencing September 2, 2018 to week commencing November 18, 2018) and a random effects term for matching group. To compare the effect of the intervention across affluence groups, interaction terms (affluence groups * intervention) were included in the model: $P_{\text{interaction}}$ for meat products = 0·738; $P_{\text{interaction}}$ for meat-free products = 0.021.

Post hoc observational analysis of the Phase I intervention by store affluence showed that changes in sales of meat products (units per store per week) were not significantly different in the intervention stores compared to the control stores regardless of the store affluence ($P_{\text{interaction}}$ = 0·738, Fig 4). However, among stores of average or below average affluence, sales of equivalent meat-free products increased markedly in the intervention stores (approximately +46%) compared to the control stores (+5%, IRR 1.61 [95% CI 1.45 to 1.78]), while this increase was less pronounced among stores of above average affluence (intervention stores (+15%) and control stores (+7% IRR 1.29 [95% CI 1.08 to 1.56]); $P_{\text{interaction}}$ = 0·021, Fig 4).

## Fidelity evaluations

The intervention was implemented in all 20 Phase I stores as planned, with 62% of them having at least 80% of items in the meat-free planogram in stock. Of the 20 stores, 15 stores had at least 1 meat-free product missing (no product shelf label), with 1 store missing 10 meat-free products. Point of sale displays were present in 13 stores. One Phase II intervention store did not add a second meat-free bay, and all other stores had considerable variation in the number and type, of meat-free products stocked within the additional bay over the Phase II intervention period. See Table H in S3 Appendix for a detailed breakdown of the fidelity evaluation results.

## Discussion

This non-randomised controlled intervention study in a large UK supermarket chain provided evidence that prominent positioning of meat-free products in the meat aisle did not result in decreased sales of equivalent meat products (the primary outcome measure). However, the

intervention led to a significant increase in sales of meat-free products, which was sustained over time. Intensification of the intervention in the Phase II period further increased sales of meat-free products, with no impact on sales of meat products. Our post hoc analysis by store affluence suggested that the effects on sales of meat-free products were of greater magnitude in stores located in areas of average or below average affluence.

There is limited and mixed evidence on the effectiveness of interventions to reduce the demand for meat where meat-free items are made more prominent [20,21], but no previous studies have tested this in real supermarket environments. Our analysis did not detect a significant effect on sales of meat products, despite the fact that the meat-free section replaced a bay where meat was previously sold in most of the intervention stores. However, sales of meat were clearly in decline over the preceding 18 months in both intervention and control stores, which may have limited the potential for the intervention to further reduce meat sales. This is consistent with declining trends in consumption of red and processed meat observed in the UK population [22,23]. Additionally, the subset of meat products studied here included a combination of fresh processed and unprocessed products (i.e., mince, burgers, meatballs, and sausages), and we cannot rule out the possibility that other sources of meat, such as carcass cuts, would have changed as a result of the intervention, and we assessed sales only of refrigerated and not frozen meat and meat-free products.

Prominent positioning of meat-free products had a significant long-lasting effect on sales of these products. Recent systematic reviews support the value of positioning strategies in retail environments to improve the nutritional quality of food purchases [13,14]. Evidence suggests that there is a greater likelihood of change when more options are available [24], and repeated exposure can also elicit increased acceptability or establish a new social norm about the presence of new products [25]. However, this study reinforces previous evidence that these nudges to encourage sales of specific products may not be sufficient to displace choices where the goal is to decrease purchases [16,26]. In practice, the range of meat-free products that were repositioned in the meat aisle had similar culinary properties and use to their equivalent meat products, and we hypothesised that they would act as direct substitutes for meat. However, the perceived palatability, price, level of processing, or acceptability of the meat-free alternatives by the purchaser or other household members may be a barrier to this substitution [27,28]. Our results provide preliminary evidence that the intervention has potential to be particularly effective in lower and middle affluence populations, which has been observed with similar policies to improve nutrition [29], although evidence supporting the potential of this kind of intervention to reduce inequalities is insufficient [14].

A strength of this study is the use of a real-world setting and objective sales data to evaluate the intervention in a natural experiment, where consumers are not aware of the intervention being tested, and, therefore, better reflects actual rather than intended behaviour. Although stores were not randomised to intervention and control conditions, we showed that they were well matched. The choice of multiple control stores with matching store sales and area characteristics (plus adjustment for confounding) may result in less biased estimates of effect. We analysed data over a long time period before and after the intervention was implemented, which allowed us to control for previous trends and seasonal changes and to explore sustained effects after the intervention. Our retail partner is a national supermarket chain with one of the largest shares of the grocery sector, and its offerings are targeted to a general and broad customer base, which increases the generalisability of the findings to the broader UK population. This study also demonstrates how academic collaborations with the food industry can yield important information to inform public health interventions.

Some methodological limitations are inherent to natural field experiments. In-store environments are inundated with multiple cues that influence food choices and are especially

challenging to control for in commercial environments. For example, stores could have implemented price promotions on meat products while positioning the meat-free alternatives, although due to national pricing policies, we would expect these to affect both intervention and control stores to a similar extent. We were also unable to look fully at substitution effects because we had data for only a limited range of products, which were part of the intervention or hypothesised as substitutes or as control products. For example, participants may have changed purchases of prime meat cuts in the same aisle or of frozen meat or meat alternatives located elsewhere in the store. We were not able to access any information regarding availability of products to understand if there were changes in the proportion of meat-free and meat products available on display, which could also explain the results. We also relied on the store implementation plans, and our fidelity evaluations revealed some deviations from the original plan in regards to signposting and positioning within the bays. However, the effect of these deviations was to reduce the likely effectiveness of the intervention.

Dietary change at the population level focusing on reductions in saturated fat, free sugars, or salt or to increase fruit and vegetable intake to reduce the risk of noncommunicable diseases is proving to be slow and especially challenging when using standard policy strategies aiming at targeting conscious decision-making processes of human behaviour [30]. There is less evidence pertaining to specific interventions to reduce meat or meet sustainability goals, although it has been suggested that physical microenvironments or social norms may override conscious intentions to follow better food practices [31,32] and that redesigning microenvironments could help shift habitual behaviours to reduce the demand for meat [8,33,34]. However, our research suggests that prominent positioning of meat-free products next to equivalent meat-products is not enough to reduce meat purchases. Positioning interventions aim at guiding food choices rather than reducing personal autonomy through structural changes. These kinds of interventions have high acceptability to governments, since they have no cost to the state, to retailers since they focus on enhancing rather than limiting sales, and the public who favour nudging over price interventions [35]. However, their limited effectiveness in meeting the primary goal, reduced meat consumption, suggests that it may be necessary to make structural changes that explicitly target meat purchases, for example, to reduce the space dedicated to meat or remove incentives to purchase meat to achieve global targets that are compatible with human health and environmental sustainability [36].

In conclusion, an intervention to make meat-free products more prominent by locating them in the meat aisle next to equivalent meat products was not effective in reducing sales of meat products, but successfully increased sales of meat-free alternatives in the long term. This research contributes to the under-explored field of effective in-store interventions to change food purchasing patterns whether to meet health or sustainability goals and supports prior evidence that positioning interventions can increase sales of prominently placed products within supermarket contexts. However, further research is needed into effective, scalable, and sustainable approaches that can reduce the demand for meat and encourage more sustainable patterns of consumption.

**Disclaimers:** The views expressed in this publication are those of the authors and not necessarily those of the National Institute for Health Research or the Department of Health and Social Care.

## Supporting information

**S1 Appendix. Study protocol.**
(PDF)

**S2 Appendix. Study statistical analysis plan.**
(PDF)

**S3 Appendix. Supporting information tables and figures.**
(DOCX)

## Author Contributions

**Conceptualization:** Brian Cook, Jennifer Hollowell, Susan A. Jebb.

**Data curation:** Carmen Piernas, Richard Stevens, Cristina Stewart.

**Formal analysis:** Carmen Piernas, Richard Stevens, Cristina Stewart, Peter Scarborough.

**Funding acquisition:** Susan A. Jebb.

**Investigation:** Carmen Piernas.

**Methodology:** Carmen Piernas, Brian Cook, Richard Stevens, Cristina Stewart, Jennifer Hollowell, Peter Scarborough, Susan A. Jebb.

**Project administration:** Carmen Piernas, Brian Cook, Cristina Stewart, Susan A. Jebb.

**Resources:** Carmen Piernas, Brian Cook, Cristina Stewart, Susan A. Jebb.

**Software:** Carmen Piernas, Richard Stevens, Peter Scarborough.

**Supervision:** Carmen Piernas, Brian Cook, Richard Stevens, Cristina Stewart, Jennifer Hollowell, Peter Scarborough, Susan A. Jebb.

**Validation:** Cristina Stewart.

**Writing – original draft:** Carmen Piernas.

**Writing – review & editing:** Carmen Piernas, Brian Cook, Richard Stevens, Cristina Stewart, Jennifer Hollowell, Peter Scarborough, Susan A. Jebb.

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
