## [Editor Report · Decision Letter 0]

1 Feb 2021

Dear Dr Piernas, 

Thank you for submitting your manuscript entitled "Estimating the effect of moving meat-free products to the meat aisle on sales of meat and meat-free products: a non-randomised controlled intervention study in a large UK supermarket chain" for consideration by PLOS Medicine.

Your manuscript has now been evaluated by the PLOS Medicine editorial staff with relevant expertise and I am writing to let you know that we would like to send your submission out for external peer review.

Kind regards,

Raffaella

Dr Raffaella Bosurgi MSc, PhD

Executive Editor, PLOS Medicine

rbosurgi@plos.org

https://twitter.com/raffi74

Remote based in London, UK

PLOS

---

## [Decision Letter · Decision Letter 1]

22 Mar 2021

Dear Dr. Piernas,

Thank you very much for submitting your manuscript "Estimating the effect of moving meat-free products to the meat aisle on sales of meat and meat-free products: a non-randomised controlled intervention study in a large UK supermarket chain" (PMEDICINE-D-21-00493R1) for consideration at PLOS Medicine. 

Your paper was evaluated by the PLOS MED editorial team editors present (Raffaella Bosurgi, Richard Turner, Caitlin Moyer, Beryne Odeny). It was also sent to independent reviewers, including a statistical reviewer. The reviews are appended at the bottom of this email and any accompanying reviewer attachments can be seen via the link below:

[LINK]

In light of these reviews, I am afraid that we will not be able to accept the manuscript for publication in the journal in its current form, but we would like to consider a revised version that addresses the reviewers' and editors' comments. Obviously we cannot make any decision about publication until we have seen the revised manuscript and your response, and we plan to seek re-review by one or more of the reviewers. 

We expect to receive your revised manuscript by Apr 12 2021 11:59PM. Please email us (plosmedicine@plos.org) if you have any questions or concerns.

We look forward to receiving your revised manuscript. 

Sincerely,

Dr Raffaella Bosurgi, 

Executive Editor 

PLOS Medicine

plosmedicine.org

Comments from the reviewers:

Reviewer #1: This non-randomised controlled intervention study aims to evaluate whether prominent positioning of meat-free products in the meat aisle was associated with a change in weekly mean sales of meat and meat-free products.

Comments:

Did the authors undertake any sample size calculations (a priori) to provide an estimate of statistical power for the study of 20 stores with matched controls?

A randomised control design would have been more rigorous and robust, with greater strength of conclusions in terms of inferring causality and minimising the risk of confounding.

However, it is good that a matched-control design has been followed, which goes some way towards understanding what might have happened in the scenario of no intervention.

Do the 108 stores included represent the wider population of stores for this retail partner? 

"Intervention sites were all stores where there was unproductive space in the meat aisle. In other words, the meat sales did not justify the volume of inventory or number of product bays. For this study, 20 stores were selected based on the retailer's estimates of the opportunity cost of removing meat products, as well as operational considerations. 

The stores were selected to have a broad spread with regard to the following characteristics: 1) affluence (based on top 2 Acorn consumer classification categories[17] within a 10 minute drive, in combination with meat-free sales performance (assessed in the 12 week period ending 25 September 2018)); and 2) store size (large, medium, small)."

Whilst Table 1 shows little difference between the characteristics of the 20 intervention stores and their matched controls, Figure 2 (left) suggests that intervention and control groups had different purchasing behaviours pre-intervention? 

Figure 2 also shows that control stores saw a slope change, despite no intervention - what might be the reasons for this?

"The primary outcome analysis used a hierarchical negative binomial model to compare changes in weekly sales (units) of meat products sold in intervention vs control stores during the main intervention period (Phase I-February 2019-April 2019). Interrupted time series analysis was also used to evaluate the effects of the phase I intervention."

The modelling technique applied is appropriate for the data and research question. 

Can the authors please clarify whether they compared weekly sales, or changes in weekly sales? 

Seasonality is almost certainly at play in this data, as acknowledged by the authors.

It is noted that: "The preintervention baseline period and the intervention period were defined a priori with the retail partner to avoid months where meat and meat-free purchases are known to be atypical (e.g. Christmas, January, summer)." This is good practice, but does not explore whether baseline months might be different to Phase I and Phase II intervention months, due to reasons other than different products in the meat aisle (such as seasonality). 

The authors go on to say that: "In sensitivity analyses we tested the effect of controlling for seasonality and changes in purchases over January." Can the authors please present these sensitivity analyses in the supplementary material?

Given the longitudinal data available prior to intervention, this seasonality effect across a complete year could be explored? ("Data on store-level weekly sales (units purchased and £) of meat and meat-free products were obtained for August 2017 to December 2019 (123 weeks) for 108 stores (20 intervention and 88 matched control stores), resulting in 13,284 total store-weeks of data").

An alternative approach might be to compare the same months as a sensitivity analysis? i.e. Feb-Apr 2018 compared to Feb-Apr 2019? 

The authors have suitably included covariates in their analysis in an attempt to account for various potential sources of confounding that may be at play:

"These measures were included as covariates in the analyses: affluence coded as average, >average, <average; age group coded as older vs younger; ethnicity coded as White, Asian, Other; area density coded as urban, more urban, less urban. "

"To assess whether differences visible in this graph were statistically significant between intervention and control stores, we used a difference-in-difference approach, calculating the mean difference in sales (£) at each week between intervention and control group, and testing whether this time series of differences changed after intervention time using a linear regression model. We used a Chow-type test for any difference (in either intercept or slope, or both) after vs. before, and used Newey-West standard errors with lag to allow for autocorrelation in the time series" 

The authors have calculated aggregate differences between control and intervention groups, and compared the linear regressions before vs. after intervention.

This is a novel approach which attempts to draw the most from the data at hand. However, it is not necessarily the most intuitive methodology, and may be difficult to interpret given the other possible differences between control and intervention groups. 

Did the authors also consider testing for differences in linear regressions for intervention groups and control groups, before vs. after (i.e. comparing intervention to intervention, or control to control, before and after intervention)? This additional analysis may help to provide some depth and improved understanding as to what is changing over time, and how. 

"Eight of the 20 stores enhanced the intervention from August 2019 onwards (Phase II intervention) by moving the entire meat-free section into the meat aisle, which was evaluated following the same analytical methods."

Thanks to Phase II of the study, the authors now have access to a cohort of stores with a) no intervention, b) some intervention and c) full intervention. Did they consider running analyses across all three of these groups in an attempt to understand if a dose-response relationship could be measured? 

Similar to the non-randomisation for the original 20 stores which may cause a bias in the data, there may be some issue regarding the selection of these 8 stores: "These eight stores were selected by the retailer based on sufficient meat-free products in stock." What might this imply about the sample of 8 stores with enhanced intervention? Is there a bias here, that might suggest the enhanced intervention was more likely to be effective?

Figure 3 shows differences between the Phase I and enhanced intervention group before Phase II started. The Phase I stores saw a step and slope change, despite no additional intervention. Why might this be?

"A small number of control stores was initially matched to more than one intervention store. In these instances, we manually selected only one control per intervention store. Therefore, of the 20 intervention stores, nine had five matched controls, ten had four matched controls, and one had three matched control stores. "

Can the authors please clarify - each control store has been uniquely matched to one intervention store? (i.e. the 88 controls refer to 88 distinct stores)?

"A preregistered protocol (https://osf.io/qmz3a/) was completed and fully available from April 2019 before obtaining data for analyses. A more detailed statistical analysis plan was completed in January 2020 after data cleaning but before data analysis. "

Can the authors please provide these within the supplementary material?

The text of the manuscript suggests that one particular retail partner or supermarket chain is included in the analysis. It is appreciated that possibly the supermarket chain cannot be named, however, can the authors please discuss how the results seem here may or may not be applicable to other chains? (different market and clientele, pricing structures etc. may affect generalisation of the study findings?).

Of note, the title of the Supplementary Material document names the retail partner. 

Given the authors have (presumably deliberately) not specified this within the manuscript, should this title be changed for commercial confidentiality reasons?

Supplementary Figure 1 shows identical weekly sales (where the lines completely overlap)- is this correct? 

Reviewer #2: I congratulate the team for the design, delivery and analysis of this impressive experiment - this is exactly the sort of information that is needed to help understand the potential impact of food industry-led interventions to change dietary practices in the UK. This has clearly been a hugely difficult study to conduct given what seems to have been quite careful "management" of the study by your industry partner and a change to the protocol mid-way through the intervention. My comments below relate largely to the accurate and appropriate reporting of your study.

1. The very helpful online protocol calls this a "pilot non-randomised before and after study". Indeed, the protocol states that one of the secondary aims of your "pilot" study is to "to obtain data needed to design and conduct a larger, more definitive in-store intervention study". Is this the case, in which should this study be better re-titled to acknowledge its pilot nature? It's noteworthy for example that there is no sample size calculation in the manuscript (you justify this in the protocol by stating this is a pilot study).

2. The online protocol relates to the study as originally planned - and this does not entirely reflect what actually happened (i.e. Phase 2). If this was an RCT then this would of course have been a major protocol change and the protocol would have need to be update. I realise that this was not an RCT but I wonder if it would have been appropriate to update your protocol to reflect what actually happened.

3. The description of the intervention(s) could be improved to help the reader. There is currently insufficient distinction between the description of Phase 1 "to their own meat-free bay in the meat aisle (l.122-123) and Phase 2 "moving the entire bay of meat-free products in the mat aisle" (l.136-137). It's not clear what "the entire bay" consists of and how that is different from "their own meat-free bay". "The meat-free bay was the last bay in the meat aisle…" surely depends on the direction in which you access the aisle - maybe "end bay" would be an alternative? It's not clear what happened in the stores between April 2019 and July 2019 (i.e. l.132-139 needs to be re-written).

4. The reader would, I'm sure, welcome a bit more description of "Acorn categories" (l.159) to help understand more clearly the metrics used to define store characteristics.

5. This reader (at least) would like to see the statistical analysis plan. It is highlighted in the text (l.164-165) but does not seem to have been made available. This is particularly important as the analysis you present in the manuscript deviates in several areas from that in your pre-specified protocol.

6. The fidelity evaluations are clearly important not only to help interpret the study but also to aid wider implementation (i.e. lessons learnt). The protocol suggests that you planned to collect a lot of detailed information about stocking and positioning of meat-free products (including photographs) and the manuscript summarises this as "to assess whether the intervention was implemented as planned". This is critical information that allows the reader to understand more about the implementation of the intervention. The 5 lines on fidelity in the result section provide insufficient and non-systematic detail on this important issue. It's not clear how the reader is supposed to interpret a finding such as "with one store missing 10 meat-free products". Was any of the fidelity information used to inform additional sensitivity analysis?

7. Table 1 is hard to interpret as the categories have not been clearly defined.

8. The August 2017-January 2019 trend analysis (supplementary Figure 1) is interesting. I am surprised that while you have noted in the text that the intervention and control stores "followed similar trends" you did not report that there is a clear and consistent difference between the groups over the entire 17 month period i.e. the control stores sold ~1-200 more units of meat than the intervention stores. To me this suggests that the matching didn't work perfectly.

9. Some of the changes identified in meat-free product sales are tiny - for example in Phase 1, meat-free mince changed from 24 units at baseline to 27 units (while the control dropped from 25 to 24). This shows up as a big effect in Figure 1 but I would suggest you consider more carefully how this is reported. Clearly the size of the change for meat-free burgers and sausages is more substantial.

10. You need to be more consistent (or at least systematic) in the way you report the primary results that I assume come from the pre-specified Phase 1 of the study. For example, why in l.242-244 to you switch from your pre-specified primary outcome "change in weekly sales (units)" to "sales (£ per store per week)"? It is not appropriate to switch outcomes in this manner.

11. The primary outcome of the study is clearly pre-specified in the protocol and repeated in the manuscript as "change in weekly sales (units)". In your manuscript you report your primary outcome (units sold) for most of Phase 1 but then use a different outcome "£ per store per week" for Phase 1 time series analyses and all Phase 2 analysis (including the very eye-catching Figures 2 and 3). Is there a reason that you have switched away from your pre-specified primary outcome for these analyses - I could not find a justification in the manuscript and again I really do not think that it is appropriate to switch outcomes in this manner without a very solid justification. It seems to me that you should reproduce Figures 2 and 3 (and the associated text) using your pre-specified primary outcome (units) and these should be the primary results that you report in this paper.

12. The opening paragraph of your discussion that reports your primary findings is currently hard to interpret without access to analysis reporting your pre-specified primary outcome (units).

Reviewer #3: This study aimed to evaluate the effect of positioning meat-free substitutes in the same aisle as the meat products. This study makes an important contribution to the literature. There are aspects of the manuscript that require further clarification and additional details. I have also outlined some methodological/presentation suggestions for the authors to consider. 

Key points:

1. The intervention could be described in greater detail in the abstract, introduction and methods sections in terms of its relevance to existing literature. It is described as increasing the prominence of the positioning of meat-free products but it is unclear how this relates to existing literature which largely describes in-store prominence as being front of store, end-of-aisle and checkouts. The novelty of this study is the revised shelf positioning of the products but this point doesn't appear to be adequately explained in the manuscript, particularly the positioning of the meat-free products along the aisle in relation to the meat products (front, mid, end of aisle and if each meat-free alternative was co-located with the meat versions or whether the meat-free section was altogether). Also whether the positioning of the meat-free products was at eye-level, or bottom shelf or above eye-level. Detailing these positioning factors of the intervention in greater detail could help improve the interpretation of findings (i.e. why some meat-free products sold more than others and why meat product sales did not decline) and aid future intervention design. 

2. This point links to that above in terms of description of numbers of products available. Greater availability/variety of products has been shown to increase sales of these products in previous literature. It would therefore be helpful for the reader to understand whether there was an increase in the availability/variety of meat-free products as part of the intervention or whether the same number of products were available at baseline, Phase I and Phase II, and how these numbers compared to the control stores. Additionally, what were the numbers/proportion of products available (both meat-free and meat) in the intervention position (both Phases) and in the pre-intervention and the control conditions; were these the same across all intervention stores? (Lines 121-126)

3. There is no description of the control condition in the methods section. Providing these details will enable the reader to better understand the intervention design, particularly the in-store and shelf positioning, and number, of meat-free and meat products available for both intervention conditions and the control condition. A full description of the control condition (and any variation across the 88 stores) should be added to the methods section. Also where were intervention and control stores located?

4. There are no sample size calculations provided for the experiment. The existing body of literature suffers from a lack of description of power calculations that account for clustering at the store level and this is an essential addition for new studies in this field, particularly for sizeable studies such as this one. It would also be helpful if the authors expanded on the reason for varying numbers of control stores; this is alluded to as a strength of the study in the discussion but no reference or clear rationale is provided.

Abstract/Introduction: 

5. Line 74-78: These sentences could be more specific in terms of my key points 1 and 2 above and how this study relates to existing literature on store/shelf prominence and the marketing practices used by supermarkets. The reference to co-location of complementary products seems fundamental to the intervention design and it would be helpful if this was more clearly explained and referenced by similar studies (e.g. Foster 2014 AJCN and de Wijk 2016 Plos One - probably others from marketing literature). 

Methods: 

6. Line 94 - it would be helpful to additional details about the store with two weeks of missing data. Was it intervention/control store? When did the two weeks fall in terms of seasons and the intervention implementation? There is great variability in retail sales weeks so taking the week before and after may not always be the best approach. If it was a control store - would have imputing from an average of the other control stores been an alternative approach? 

7. Lines 104-108; lines 158-161 - The authors conclusions about potential impact on inequalities is based on a measure that of socioeconomic status (SES) that may be unfamiliar to journal readers. Additional details about the affluence measure and how it relates to more typical measures of SES used by researchers would improve understanding and relevance of the associated findings. Additionally, the descriptions of the other variables to determining store matching and confounding variables could be improved to enable clear interpretation of these variables. (i.e. was is meant by meat-free sales performance - total, % sales, alternative? Can examples of store size be provided? What is average affluence? What is the cut-off for young and old? What is meant by average urban, more or less urban?)

8. Lines 109-117 - Use of a computerised matching system provides an objective methods for matching which is beneficial - though reliant on the supermarket rather than the research team. It is difficult to understand how the research matching linked to this matching through the supermarket store details system. Could this description be revisited and a rationale provided. 

9. Lines 140-146 - would be better placed in another sub-section as they don't directly relate to the intervention description. 

10. Related to key-points 1 and 2, it would be helpful if sensitivity analyses were conducted to assess differences in changes to availability of meat-free and meat products across intervention/control stores. (i.e. lines 123-125 state that meat-free replaced meat products suggesting potential decrease in meat product availability and if proportions available altered across stores this additional information about differential intervention effects would make a useful contribution to existing literature).

11. Lines 205-208 - could further details about the measures used to assess fidelity be outlined in this section and a table of the assessments be provided in the results section/supplementary materials? These additional details again provide a helpful addition to previous work. 

Results: 

12. Table 1 - could size of store be added here too? It would be interesting to view the differences in store size across the study stores.

13. Suggest including Supplementary Table 3 in the main body of the results because this contains important detailed information.

14. Figures 2 and 3 seem to be missing a key - difficult to interpret without this when in black and white print. Also, it would be helpful to see the longer-term effect of the intervention in the interrupted time series analyses. Could you test the effect again at time point Nov 2019 not just at the time of implementation? 

Discussion:

15. Related to key-points 1 and 2, it would be helpful if the discussion reflected the potential differential effects of positioning and availability and how this relates to existing literature. 

16. Lines 39-42 (page 20) these analyses are important to have conducted but the interpretation here seems a little inflated because low affluence stores were combined with mid affluence and these were taken at the store rather than household/individual level. I would recommend revising this sentence in light of these factors. 

References:

17. References 17 and 18 appear to be the same reference. It would be beneficial for the reader if additional access details were added (i.e. weblink and date accessed if online or full details of author and publisher of only available in print).

[LINK]

---

## [Editor Report · Decision Letter 2]

24 Jun 2021

Dear Dr. Piernas,

Thank you very much for re-submitting your manuscript "Estimating the effect of moving meat-free products to the meat aisle on sales of meat and meat-free products: a non-randomised controlled intervention study in a large UK supermarket chain" (PMEDICINE-D-21-00493R2) for review by PLOS Medicine.

I have discussed the paper with my colleagues and the academic editor-Alan Dangour I am pleased to say that provided the remaining point raised by Alan Dangour and editorial/production issues are dealt with we are planning to accept the paper for publication in the journal.

Aland Dangour: For total clarity and scientific transparency I would like to make one further request. They have (as is good practice) noted with dates the amendments/updates they have made to their protocol (following our earlier review comments). However they have not done this for their statistical analysis plan (SAP). Given the importance of the SAP I think it would be appropriate for them to note (with dates) the changes they have made to this document during the latter stages of the study.

[LINK]

We look forward to receiving the revised manuscript by Jul 01 2021 11:59PM.   

Sincerely,

Dr Raffaella Bosurgi, 

Executive Editor

PLOS Medicine

plosmedicine.org

Requests from Editors:

Comments from Reviewers:

[LINK]

---

## [Editor Report · Decision Letter 3]

29 Jun 2021

Dear Dr Piernas, 

On behalf of my colleagues and the Academic Editor, Alan Dangour , I am pleased to inform you that we have agreed to publish your manuscript "Estimating the effect of moving meat-free products to the meat aisle on sales of meat and meat-free products: a non-randomised controlled intervention study in a large UK supermarket chain" (PMEDICINE-D-21-00493R3) in PLOS Medicine.

PRESS

Sincerely, 

Dr Raffaella Bosurgi 

Executive Editor

PLOS Medicine